# Comparing performance of primary care clinicians in the interpretation of SPIROmetry with or without Artificial Intelligence Decision support software (SPIRO-AID): a protocol for a randomised controlled trial

Gillian Doe [1] , Ethaar El-Emir,[2] George D Edwards,[2] Marko Topalovic,[3] Rachael A Evans [1] , Richard Russell,[4] Karl P Sylvester,[5] Karolien Van Orshoven,[3] Anthony P Sunjaya,[2,6] David A Scott,[7] A Toby Prevost [8] , Jennifer Harvey,[2] Stephanie JC Taylor,[9] Nicholas S Hopkinson,[10] Samantha S Kon,[2,11] Ian Jarrold,[12] Nannette Spain,[13] Winston Banya,[2,14] William D-C Man[2,4,15]

For numbered affiliations see end of article.

**Correspondence to**
Dr Gillian Doe;
ged6@leicester.ac.uk

## ABSTRACT

**Introduction** Spirometry is a point-of-care lung function test that helps support the diagnosis and monitoring of chronic lung disease. The quality and interpretation accuracy of spirometry is variable in primary care. This study aims to evaluate whether artificial intelligence (AI) decision support software improves the performance of primary care clinicians in the interpretation of spirometry, against reference standard (expert interpretation).

**Methods and analysis** A parallel, two-group, statistician-blinded, randomised controlled trial of primary care clinicians in the UK, who refer for, or interpret, spirometry. People with specialist training in respiratory medicine to consultant level were excluded. A minimum target of 228 primary care clinician participants will be randomised with a 1:1 allocation to assess fifty de-identified, real-world patient spirometry sessions through an online platform either with (intervention group) or without (control group) AI decision support software report. Outcomes will cover primary care clinicians' spirometry interpretation performance including measures of technical quality assessment, spirometry pattern recognition and diagnostic prediction, compared with reference standard. Clinicians' self-rated confidence in spirometry interpretation will also be evaluated. The primary outcome is the proportion of the 50 spirometry sessions where the participant's preferred diagnosis matches the reference diagnosis. Unpaired t-tests and analysis of covariance will be used to estimate the difference in primary outcome between intervention and control groups.

**Ethics and dissemination** This study has been reviewed and given favourable opinion by Health Research Authority Wales (reference: 22/HRA/5023). Results will be submitted for publication in peer-reviewed journals, presented at relevant national and international conferences, disseminated through social media, patient and public routes and directly shared with stakeholders.

**Trial registration number** NCT05933694.

## STRENGTHS AND LIMITATIONS OF THIS STUDY

⇒ Real-world spirometry traces completed in primary care, irrespective of technical quality, will be used.
⇒ To replicate real-world primary care in the UK, study participants will not be limited to general practitioners, but will include other members of multidisciplinary team that are expected to perform or/and interpret spirometry in primary care.
⇒ The trial will be advertised and recruited nationally to maximise participation and variation in participant sample.
⇒ The trial statistician will be blinded to group allocation.
⇒ Trial outcomes will include all relevant aspects of spirometry interpretation, including assessment of technical quality, pattern recognition and diagnosis prediction.

## INTRODUCTION

Chronic respiratory disease has significant negative impact on quality of life and is the third leading cause of death globally, and one of the highest contributors to economic burden for healthcare systems.[1] Chronic obstructive pulmonary disease (COPD) alone accounted for 3.3 million deaths worldwide in 2019, increasing 14% over the previous 10 years.[2] Accurate and timely diagnosis of respiratory disease is key to improving access to treatment and patient outcomes, and long delays to diagnosis for individuals with chronic lung diseases are well documented.[3–5]

Spirometry is a simple, point-of-care lung function procedure recommended to support

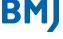

the diagnosis and monitoring of COPD and asthma, the most common long-term respiratory conditions.[6 7] In many western and low/middle-income countries, spirometry will be performed and interpreted in non-specialist respiratory settings. However, there is inequity in spirometry provision in primary care and community settings; historical challenges with quality, skills and competence in interpretation, and funding, have been compounded by the COVID-19 pandemic.[8] Spirometry performed in primary care and community settings has been shown to not meet international criteria in over 85% of cases.[9] A low level of confidence in assessing spirometry quality is also common among general practitioners (GPs).[10] Improving prevention and early diagnosis has been identified as a priority in the Global Impact of Respiratory Disease report.[11] In practice, this requires improved equity, access and quality of spirometry provision.

Recent work has shown that artificial intelligence (AI)-powered decision support software can provide automated technical quality assessment and interpretation of full lung function tests.[12 13] This software can outperform pulmonologists in diagnostic prediction from lung function tests.[13] Similar AI software that interprets spirometry (usually the only lung function test accessible in many healthcare settings) may be particularly helpful in primary care to support clinicians who have generalist roles and less experience in respiratory diagnosis. Qualitative work highlighted that validation and evaluation of the effectiveness of AI software is key in engaging clinicians and commissioners to have confidence in implementation in clinical practice.[8]

The primary objective of this study is to compare the performance of primary care clinicians in the interpretation of 50 real-world spirometry records with or without AI spirometry decision support software reports.

## METHODS AND ANALYSIS
### Trial design and registration
This is a parallel, two group, randomised controlled, statistician-blinded efficacy trial to evaluate whether AI spirometry decision support software can improve the performance of primary care clinicians in their interpretation of real-world spirometry. Each clinician will be provided with the same clinical dataset of 50 de-identified, real-world patient spirometry records through an online platform (Qualtrics XM), and asked to answer questions about preferred diagnosis (ie, diagnosis they think is most likely) and differential diagnosis (ie, second most likely diagnosis) based on spirometry and limited clinical data, the technical quality of spirometry and the spirometry pattern. Questions will also explore clinicians' confidence in their assessments.

Participants will be allocated at random to receive either spirometry records alone or spirometry records with the addition of the AI software report. The clinical spirometry records will be de-identified (name, date of birth, address, postcode, occupation, GP, medications data removed), by a member of the clinical care team.

Royal Brompton and Harefield Hospitals (RBHH), Guy's and St Thomas' NHS Foundation Trust (GSTFT) will act as study sponsor. This study has been reviewed and given favourable opinion by Health Research Authority (HRA) Wales (reference: 22/HRA/5023).

### Study population
Potential participants will be clinicians working in primary care involved in spirometry referral and interpretation.

### Eligibility criteria
Eligibility criteria will be: (1) a clinician working in primary care (defined as >50% of job plan in primary care) in the UK who refers for or performs spirometry (typically GP, practice nurse); (2) able to access an online study platform in order to review spirometry sessions; (3) able to provide informed consent via the study platform.

Exclusion criterion will be: (1) clinicians who have completed specialist training in respiratory medicine and are recognised by the General Medical Council with the right to practise as a consultant in respiratory medicine in the National Health Service.

Eligibility criteria will be used as screening questions on the online study platform prior to participants being able to consent to participate.

Participants will be randomised 1:1 to either control (report 50 spirometry records alone) or intervention (report same 50 spirometry records with access to an AI spirometry interpretation report for each spirometry result). Randomisation will be done by the study online platform at the point of consent to the study by minimising according to job role (GP: yes or no) and spirometry accreditation from the Association for Respiratory Technology and Physiology (performance only, interpretation only, performance and interpretation, no accreditation) and then randomising by a random number generation algorithm performed by the online platform.

### Recruitment and consent
Recruitment to the study will be advertised through national and local primary care networks, professional networks, the institution website and social media will be used. By advertising and recruiting nationally, the recruitment strategy aims to engage with clinicians from diverse backgrounds in terms of the patient population they serve, age, years of clinical experience, job role and ethnicity. We will offer financial reimbursement for all potential participants participating and completing the study, which may attract some who might not be interested in AI or respiratory diagnostics.

Clinicians will be sent a letter outlining the research study with the option for them to express interest in participating. The letter serves as the participant information sheet (see online supplemental information) and will be sent via email. Participants who express interest in taking part will be sent a link to the online study

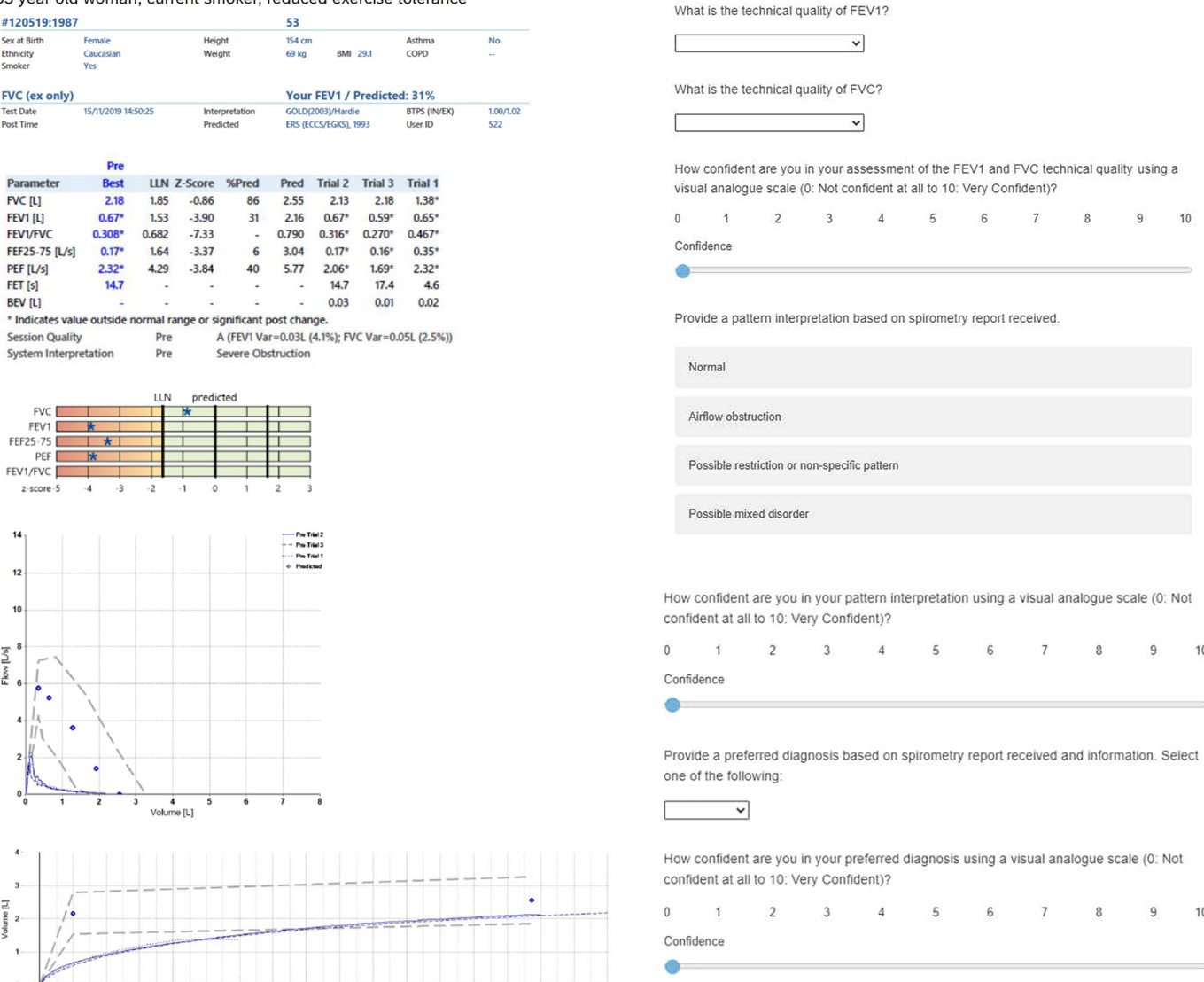

**Figure 1** Example spirometry case and questions for primary care clinicians. FEV$_1$, forced expiratory volume in 1 s; FVC, forced vital capacity.

platform where they will be able to complete the eligibility screening questions and, if applicable, consent form online. The study will also be advertised on the RBHH research website with a link available to the online study platform. Participants will be able to withdraw without giving any reason, and the number of participants registering but not completing the study will be recorded. Randomisation will be kept concealed until the point of the allocation of the participant to a study arm.

## Setting
The study will be conducted via an online platform (Qualtrics XM).

## Procedure
Study participants (participating clinicians) will independently assess the same 50 spirometry records, in the same order, through a bespoke designed online platform developed using Qualtrics XM (Seattle, USA).

Participants will be blinded to the reference standards. See figure 1 for example. The patient spirometry records were randomly selected from a database comprising 1122 patients undergoing spirometry in primary care and community-based respiratory clinics in Hillingdon borough between 2015 and 2019. An online randomiser will be used to select the 50 spirometry sessions used in the trial (https://www.randomizer.org/), according to 'disease category' so that of the 50 spirometry traces selected, 40% will be from patients with COPD, 20% will have normal spirometry and 10% for each of the four other disease categories (asthma, interstitial lung disease (ILD), other obstructive, other disease/unidentifiable category), to represent a likely distribution of diagnoses in a primary care setting.

For each spirometry record, the primary care clinician participant will answer the following questions on the Qualtrics platform:

► What is the technical quality of the spirometry? (a drop-down box containing the options: acceptable (quality grade A or B), not acceptable). Technical quality will be defined according to the ATS/ERS 2019 Spirometry Technical Statement, provided to participants via a link embedded in the online platform.

  – One response for forced expiratory volume in 1 second ($FEV_1$) and one question for forced vital capacity (FVC).

► How confident are you in your technical quality assessment using a visual analogue scale (0: not confident at all to 10: very confident)?

► Provide a pattern interpretation based on spirometry report received. Options: normal, airflow obstruction, possible restriction or non-specific pattern, possible mixed disorder.

► How confident are you in your pattern interpretation using a visual analogue scale (0: not confident at all to 10: very confident)?

► Provide a preferred diagnosis based on spirometry report received and information (a drop-down box containing the options: asthma, COPD, ILD, normal lung function, other obstructive disease and other unidentifiable disease): **preferred diagnosis**.

► Provide a differential diagnosis based on spirometry report received and information (a drop-down box containing the options: asthma, COPD, ILD, normal lung function, other obstructive disease and other unidentifiable disease): **differential diagnosis**. The previously chosen 'preferred diagnosis' is automatically removed from the list of available options.

► How confident are you in your diagnosis using a visual analogue scale (0: not confident at all to 10: very confident).

Participants will have 8 weeks to review the 50 spirometry traces; reminders will be sent at weeks 3, 6 and 7 to those who have not completed their assessments (figure 2).

### Intervention: AI spirometry software

The AI software was developed by ArtiQ (Leuven, Belgium), and provides AI-supported quality assessment and interpretation guidance of spirometry sessions (online supplemental file). The quality assessment component leverages deep learning methods to perform the subjective assessment of spirometry quality assessment (ie, related to curve shape), as well as implementing the objective criteria from international guidance (ie, related to numeric criteria). The AI component of the software mimics the subjective visual inspection of data usually performed by technicians. Per session it provides an overall session quality grade according to ATS/ERS 2019 Spirometry Technical Statement (A–F) and calculates the best trial (spirometry trace) which should be considered for diagnostic interpretation. The model was trained based on spirometry measurements from the National Health and Nutritional Examination Survey (NHANES 2011–2012)[12] and validated in clinical trial settings.[14]

The spirometry interpretation component is a decision support software focusing on the diagnostic interpretation of spirometry sessions. It provides support in pattern interpretation according to international interpretation guidelines (ie, normal, obstructive and/or restrictive)[15] and a disease probability. The disease probability model was originally trained to distinguish eight of the most common categories (seven diseases+normal lung function) detectable with full pulmonary function testing.[13] For use in primary care settings, the model has been adapted to identify six categories—asthma, COPD, ILD, normal lung function, other obstructive disease (such as

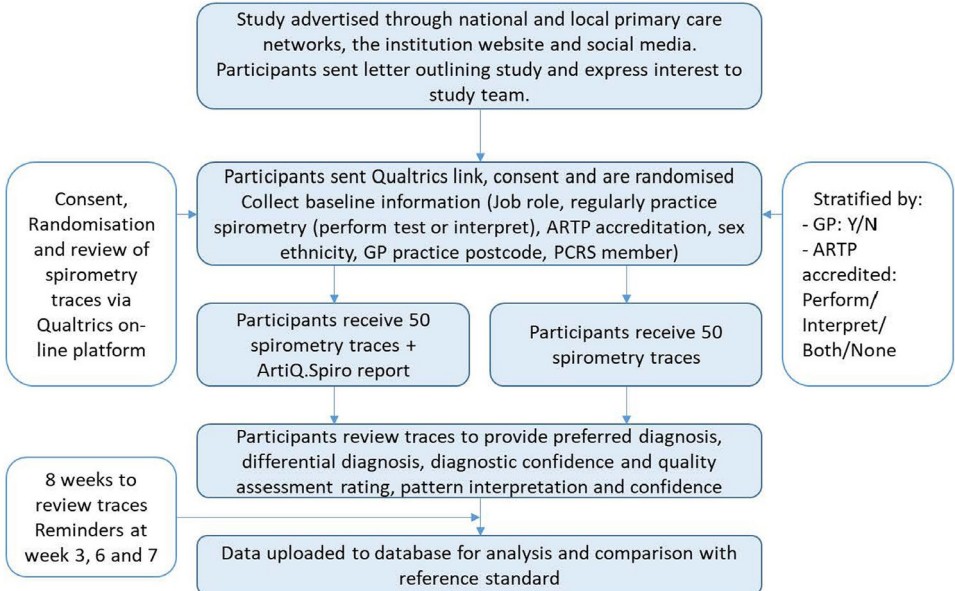

**Figure 2** Study schedule. ARTP, Association for Respiratory Technology and Physiology; GP, general practitioner.

cystic fibrosis and bronchiectasis) and other unidentifiable disease—solely based on spirometry using the same training set. The rationale for reducing the number of categories to identify is based on the reduced lung function dataset available when working in a primary care environment compared with a hospital-based pulmonary function testing laboratory. The categories selected are those most commonly found in primary care, with spirometry testing indicated mainly in COPD and asthma diagnosis.

When the quality model identifies the data quality of the spirometry session to be poor (ie, quality of $FEV_1$ and/or FVC is not A or B according to ATS/ERS 2019 guidelines), the following disclaimer is added to the interpretation report: 'Data quality is not sufficient. Interpretation above might not represent the patient's lung function'. Participants will still be asked for a diagnosis and spirometry interpretation despite the bad quality.

### Reference standard

Reference standards for diagnosis for the 50 spirometry traces were defined before commencement of the trial, by two pulmonologists. Primary and secondary care medical records were reviewed independently by two pulmonologists from the RBHH, UK. The diagnostic reference standard was attributed to one of six categories (COPD, asthma, ILD, other obstructive disease, normal, other) by each pulmonologist. In participants with multiple respiratory diagnoses (eg, COPD and ILD), the pulmonologist was asked to choose, to the best of their abilities, the predominant category. If consensus could not be reached between the two pulmonologists, these cases and their medical records were reviewed by a third pulmonologist to adjudicate independently. All pulmonologists (SSK, WD-CM, NSH) were accredited specialists in respiratory medicine with a minimum of 8 years as a consultant in the National Health Service, with expertise in the diagnosis and management of COPD and other chronic respiratory diseases. Pulmonologists had no access to the AI software reports, nor had any communication with the software engineer or software company.

Reference standards for the technical and pattern interpretation were completed by a respiratory physiologist (KPS) with 26 years experience. The respiratory physiologist was asked to rate the technical quality of the $FEV_1$ and FVC using the categories acceptable (quality grade A and B ATS/ERS 2019 guidelines) or not acceptable (quality grade C, D, E, F, U).[15] For pattern interpretation of the spirometry, the respiratory physiologist was asked to choose from the following four categories: normal, airflow obstruction, possible restriction or non-specific pattern, possible mixed disorder.

All pulmonologists and respiratory physiologist were blinded to AI software reports.

### Study outcomes

The primary endpoint for a primary care clinician participant is the performance of their preferred diagnosis, defined to be the number of correct cases identified expressed as a percentage out of the total number of spirometry records. A correct case is where the participant's preferred diagnosis matches the reference final diagnosis (described earlier). The secondary endpoints, expressed per participant are listed in table 1.

**Table 1** Secondary endpoints

| Secondary endpoints | Definition | Measured | Reference standard |
|---|---|---|---|
| Quality assessment of the spirometry trace | A correct case is where the participant's quality grade matches the reference quality grade. Units will be percentage of total cases that are correct | Multiple choice categories: acceptable (grade A/B) or not acceptable (grades C/D/E/F/U) | Respiratory physiologist |
| Quality assessment self-rated confidence | | Visual analogue scale (0–10) where is 0=not confident at all; 10=very confident | |
| Pattern interpretation of the trace | A correct case is where the participants' selected pattern matches the reference pattern. Units will be percentage of total cases that are correct | Multiple choice categories: normal, airflow obstruction, possible restriction or non-specific pattern, possible mixed disorder | Respiratory physiologist |
| Pattern interpretation self-rated confidence | | Visual analogue scale (0–10) where is 0=not confident at all; 10=very confident | |
| Differential diagnostic performance | A correct case is where the preferred or differential diagnosis matches the reference final diagnosis. Units will be percentage of total cases that are correct | Multiple choice categories: asthma, chronic obstructive pulmonary disease, interstitial lung disease, normal lung function, other obstructive disease and other unidentifiable disease | Pulmonologist |
| Diagnostic self-rated confidence | | Visual analogue scale (0–10) where is 0=not confident at all; 10=very confident | |

## Sample size calculation

The sample size calculation was informed by previous feasibility data from 30 primary care practitioners assessing five spirometry traces. The preferred diagnosis matched the reference diagnosis in a mean (SD) of 55% (19%) of spirometry records. We have assumed the same SD of 19%, which is conservative because more cases will be assessed per practitioner, leading to less variable practitioner rates. As the calculation involves conservative assumptions, the power was set below 90%, to be 85%. With 132 participants (66 per group) this allows for detection of a 10% difference (a mean of 5 extra cases in 50) in the rate of cases correctly identified to match the reference standard (mean 65% vs mean 55%) using an unpaired t-test at the two-sided 5% significance level.

The recruitment target was originally estimated at 156 practitioners to allow for an estimated 15% drop out. However, a preliminary analysis of completion rate of the first 50 recruited participants demonstrated that 29 of the 50 participants scored all 50 spirometry records within the 8-week study period (58% completion). Based on this data, the study statistician advised increasing the recruitment target to 228 to account for 42% non-completion (see the Amendment section).

Descriptive statistics will be performed for demographics, baseline characteristics and endpoint data: number and percentage for categorical data and mean and SD or median and interquartile range (IQR) for non-normally distributed continuous data.

## Statistical analysis

Survey responses from participants will be collated in a secured, online database. A Statistical Analysis Plan will be developed and statistical analysis will be conducted by the study statistician. The statistician will be blinded to control/intervention group allocation. Descriptive statistics will be presented for demographic, baseline characteristics and endpoint data. Number and percentage will be presented for categorical data. Mean and SD will be presented for normally distributed continuous data while median IQR will be presented for non-normally distributed continuous data.

The number of participants consented, randomised, and completing the study will be described in a study Consolidated Standards of Reporting Trials diagram.

Data analysis will be performed on an intention to treat basis. Data will be analysed with the statistical software Stata V.17.0. All statistical tests will be two-sided and significance set at $p < 0.05$.

### Primary endpoint analysis

For each participant, diagnostic performance will be calculated as percentage of the 50 spirometry procedures where the final diagnosis was correctly identified. In each randomised group, the diagnostic performance will be summarised as the mean of the participants' diagnostic performance. A two-sample unpaired t-test will be used to assess the mean difference in correct response between

the two randomised groups. The assumptions for the two-sample t-test will be tested and where the assumption for equal variances fails then the Welch t-test will be used.

Randomisation will be stratified according to job role (GP: yes or no) and Accreditation from Association for Respiratory Technology and Physiology (performance only, interpretation only, performance and interpretation, no accreditation), therefore an analysis of covariance will be used to compare the intervention and control groups by adjusting for the main effects of the stratified variables. To internally validate the estimate of the difference between the two groups bootstrap estimation of difference will be done by taking 100 from the analysis of covariance model or Welch's t-test.

### Secondary endpoint analysis

For continuous variables, the secondary endpoints will be compared between the control and intervention groups using the same approach as the primary endpoint.

The diagnostic prediction performance of the AI software alone compared with the reference diagnosis by the pulmonologists will also be reported.

### Subgroup analysis

Subgroup analysis will be performed on the 20 spirometry traces where the confirmed diagnosis (reference standard) was COPD, as spirometry in primary care is most relevant to COPD diagnosis.

### Missing data

Only participants who complete at least 70% of the spirometry cases will be included in the primary analysis. We will compare the baseline characteristics and demographics of those who completed <70% of questions with those who completed ≥70% of questions to identify any systematic differences between completers and non-completers.

## Data collection

Data will be collected via the online platform Qualtrics which is general data protection regulations (GDPR) compliant. Further information is provided online (https://www.qualtrics.com/uk/platform/gdpr/).

A user license has been obtained by RBHH to use this platform for research purposes. There will be one secure user log in for one member of the study team who will be unblinded. All other members of the research team will be blinded.

## Data management

All data will be handled in accordance with the Data Protection Act (2018), NHS Caldecott Principles, The UK Policy Framework for Health and Social Care Research, and the condition of the REC approval. The online platform (Qualtrics) will be used to collect data responses from each participant reviewing the spirometry traces. Qualtrics itself will act as a database and data will also be downloaded to an excel file/csv file/database) for back up and analysis. Database access will be strictly restricted through user-specific passwords to the authorised research

team members. All returned data (Microsoft Forms and Excel spreadsheets) will be held on the H drive (Harefield Hospital) by the clinical team (CI and study team) and will be password-protected and only accessible by the direct study team.

No participant identifiable data, beyond basic demographics, will be transferred and all participants will have a participant identification number. Data will be downloaded typically within 1 week of data collection on Qualtrics by members of the research team. The research team will undertake appropriate reviews of the entered data where appropriate for the purpose of data cleaning and will request amendments as required.

At the end of the study, the CI or their assigned delegate will review all the data for each participant to verify that all the data are complete and correct. At this point, all data can be formally locked for analysis.

### Patient and public involvement

Patient representatives are included in the trial management group and will attend the quarterly meetings across the duration of the trial. The Deputy Head of Research and Innovation for Asthma+Lung UK (the largest UK lung health charity) is also part of the trial management group. Our research team includes two costed coapplicants who will colead the patient and public involvement (PPI) elements of the project.

PPI coapplicant responsibilities include: setting and refining overall PPI strategy as the project progresses, reporting on PPI activities to the research management group, evaluation, monitoring and reporting of PPI for example, using a PPI impact log, communicating with our PPI collaborators (Asthma+Lung UK) as well end-users (GP federations/clinical commissioning groups) on the project status, synthesising results and conclusions of PPI activities and disseminating feedback, writing up PPI sections with public contributors for ethics applications, patient-facing documents and project reports.

### Safety reporting

The procedure proposed for this efficacy study will not affect the usual standard of care for participants or patients. The datasets will comprise spirometry records previously collected as part of clinical spirometry pathways in primary care. As such, this retrospective analysis, which will not involve delivery of an intervention nor a change in patient's usual clinical care, is unlikely to produce direct risk for participants.

The main study risks involve patient confidentiality, information governance and avoiding bias. To address this, all data analysed by parties outside the direct clinical team will be de-identified. Spirometry records will be anonymised and provided to clinicians on an online platform (Qualtrics) compliant with GDPR.

The research team (except the platform administrator) and trial statistician will be blinded to participants' group allocation until the completion of data analysis for the primary and secondary outcomes.

### Protocol amendments

Any changes to the study protocol outlined in this paper will be approved by HRA and study Sponsor RBHH, GSTFT. Two non-substantial amendments have been submitted and approved by HRA for this study: (1) addition of a link to the study platform in the participant information; (2) an increase in target recruitment number (detailed in the Methods section).

## ETHICS AND DISSEMINATION
### Ethical approval

This study has been reviewed and given favourable opinion by HRA Wales (reference: 22/HRA/5023).

### Dissemination

To maximise the audience for dissemination of our findings, results from the study will be disseminated by presentations at relevant scientific meetings and conferences as well as by high impact peer-reviewed publications. We will also disseminate via presentations and newsletters to participants and key stakeholders, including Asthma+Lung UK who have been collaborators on this grant. The results will be shared with local and national primary and secondary care partners and networks, including British Thoracic Society, Primary Care Respiratory Society and International Primary Care Respiratory Group. A summary report will be shared on the Royal Brompton and Harefield website research pages.

## DISCUSSION

Spirometry is a key investigation for supporting diagnosis and disease monitoring in chronic respiratory disease. There is a clear need to improve the access, quality and interpretation of spirometry in primary care and other non-respiratory community settings which have many historic challenges, further negatively impacted by the COVID-19 pandemic.[8] AI decision-support software has shown potential in improving quality assessment and interpretation of spirometry and this study will evaluate its effectiveness in interpretation of real-world spirometry results assessed by clinicians working in primary care in the UK.

This study is the first randomised controlled efficacy study with an intervention designed to improve the performance of primary care clinicians in the interpretation of spirometry using AI and an important first step for clinicians to make informed choices about its use. It will directly test the effect of an AI-powered spirometry interpretation software in primary care using real-world traces reflective of the results reviewed in practice by clinicians.

We will advertise the trial and recruit nationally to maximise participation and variation in participant sample.

### Author affiliations
[1]NIHR Biomedical Research Centre, University of Leicester, Leicester, UK
[2]Harefield Respiratory Research Group, Royal Brompton & Harefield Hospitals, Guy's and St Thomas' NHS Foundation Trust, UK, London, UK

3ARTIQ, Leuven, Belgium
4Kings Centre for Lung Health, King's College London, London, UK
5Cambridge Respiratory Physiology, Cambridge University Hospitals NHS Foundation Trust, Cambridge, UK
6The George Institute for Global Health, Sydney, New South Wales, Australia
7Southampton Health Technology Assessments Centre, University of Southampton, Southampton, UK
8Nightingale-Saunders Clinical Trials and Epidemiology Unit, King's College London, London, UK
9Wolfson Institute of Population Health, Queen Mary University of London, London, UK
10National Heart and Lung Insititute, Imperial College London, London, UK
11Department of Respiratory Medicine, Hillingdon Hospitals NHS Foundation Trust, Uxbridge, UK
12Asthma and Lung UK, London, UK
13PPI Partner, London, UK
14Medical Statistics, Research & Development, Royal Brompton & Harefield Hospitals, Guy's and St Thomas' NHS Foundation Trust, London, UK
15National Heart & Lung Institute, Imperial College London, UK, London, UK

**Acknowledgements** We thank our patient and public involvement members for their contribution to the overall grant application and involvement in trail management meetings. We acknowledge Asthma+Lung UK for their contribution and support with this work.

**Contributors** WM and MT conceived the wider research plan. WM, MT, RAE, RR, SSK, ST, ATP, NSH, NS were coapplicants for the wider grant. All authors developed the design and plan for this study. WB, WM, GD completed the statistical analysis plan which was approved by all authors and ATP provided statistical expertise for the wider grant. MT and ArtiQ team developed the intervention. NS, IJ and NSH led the patient and public involvement. APS and GDE set up the study on the Qualtrics platform. GDE, EE-E and WM drafted the initial manuscript. All authors (GD, EE-E, GDE, MT, RAE, RR, KS, KO, APS, DAS, ATP, JH, ST, NSH, SSK, IJ, NS, WB and WM) contributed to the development of the study protocol and reviewed, commented and approved the manuscript.

**Funding** The study is being funded by National Institute for Health Research (NIHR) through an AI Award in Health and Care (Phase 3- Application: Grant number AI_AWARD02204). The study is also supported by the NIHR Leicester Biomedical Research Centre – Respiratory theme. ST is supported by the National Institute for Health Research ARC North Thames. RR is supported by the NIHR Oxford Biomedical Research Centre – Respiratory. RAE is supported by an NIHR Clinical Scientist fellowship. The views expressed in this publication are those of the authors and not necessarily those of the National Institute for Health Research or the Department of Health and Social Care.

**Competing interests** WM is Honorary President of the Association for Respiratory Technology and Physiology (ARTP). MT is CEO of ArtiQ—a company that produces artificial intelligence enabled lung function support software. JM and KO are employees of ArtiQ.

**Patient and public involvement** Patients and/or the public were involved in the design, or conduct, or reporting, or dissemination plans of this research. Refer to the Methods section for further details.

**Patient consent for publication** Not applicable.

**Provenance and peer review** Not commissioned; externally peer reviewed.

**ORCID iDs**
Gillian Doe http://orcid.org/0000-0003-4782-5811
Rachael A Evans http://orcid.org/0000-0002-1667-868X
A Toby Prevost http://orcid.org/0000-0003-1723-0796

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
