## [Reviewer comments · BMJ Open]

ARTICLE DETAILS

TITLE (PROVISIONAL)	Comparing performance of primary care clinicians in the interpretation of SPIROmetry with or without Artificial Intelligence Decision support software (SPIRO-AID): a protocol for a randomised controlled trial.
AUTHORS	Doe, Gillian; El-Emir, E; Edwards, G; Topalovic, Marko; Evans, Rachael; Russell, Richard; Sylvester, Karl; Orshoven, K; Sunjaya, A; Scott, D; Prevost, A Toby; Harvey, J; Taylor, Stephanie; Hopkinson, Nicholas; Kon, S; Jarrold, I; Spain, N; Banya, W; Man, William

VERSION 1 – REVIEW

REVIEWER	Mahesh, Padukudru JSS Medical College
REVIEW RETURNED	06-Apr-2024

GENERAL COMMENTS	It is an important study and one that will help the GPs manage their patients better in primary care. The protocol is well written and addresses comprehensively all the aspects of the proposed study It would be useful to have randomly selected the GPs from the register rather than have them answer advertisements. That would be more representative and given the real scenario. Here the sample of GPs would be biased towards those who have an interest in AI It would be useful for the GPs who don't have access in the first round to use it in the second phase and take their feedback
--

REVIEWER	Brems , J Henry Johns Hopkins University, Department of Pulmonary and Critical Care Medicine
REVIEW RETURNED	09-Apr-2024

GENERAL COMMENTS	The SPIRO-AID study is an interesting and important study to assess the effectiveness of an AI-intervention to primary care clinicians' assessments of spirometry. The study is well-designed with clearly explained rationale. My major questions relate to determinations of diagnosis, which is the key outcome being assessed here, given the existence of varying standards, importance of other PFT metrics, and importance of clinical history in making a diagnosis. 1) Do all reports only include spirometry (with no lung volumes or lung capacity), and do they only include pre-bronchodilator
---

	spirometry? Spirometry alone is already limited in its diagnostic ability. ILD is not diagnosed based on spirometry alone (and even 'restriction' is best defined based on lung volumes). COPD is preferably diagnosed on post-bronchodilator spirometry. Further, differentiating COPD and asthma would be assisted by post-bronchodilator testing or clinical history. 2) Along the same lines, what clinical history or context is provided in the cases to participants? Differentiating between many of the diagnoses frequently relies on other clinical history, including symptoms, risk factors, and radiology. 3) Were the two pulmonologists who determined the diagnostic reference standard directed to use any specific criteria? Most pertinently given the focus on COPD, there are varying recommendations on whether to use the FEV1/FVC criteria below the lower limit of normal, or below 70% predicted. 4) Additionally for the determination of the diagnostic reference standard, is there any assessment of either inter-rater reliability or how pulmonologist interpretations match the AI-generated interpretations? 5) Page 15, line 35 – Will any subgroup analysis be done of the other specified conditions? 6.) Page 15, line 41 – Is there any specific reasoning for 70%? Are there any additional planned analyses in which those with <70% completion would be included?
--	---

REVIEWER	Deepak, Desh Dr Ram Manohar Lohia Hospital and Post Graduate Institute of Medical Education and Research
REVIEW RETURNED	19-Apr-2024

GENERAL COMMENTS	It is a well designed study. This study has the potential to serve as an important link to the future where the interpretation of spirometry may be totally automated based on AI. The challenge will be to ensure that clinicians who volunteer to participate in the study do not upgrade their existing understanding regarding spirometry as that will lead to the study bias. Broad advertisement for diverse participant enrollment would be needed. The study design has is meticulous and covers all required aspects.
--

VERSION 1 – AUTHOR RESPONSE

Reviewer: 1

C2: It is an important study and one that will help the GPs manage their patients better in primary care. The protocol is well written and addresses comprehensively all the aspects of the proposed study

R2: Thank you for these positive comments.

C3: It would be useful to have randomly selected the GPs from the register rather than have them answer advertisements. That would be more representative and given the real scenario. Here the sample of GPs would be biased towards those who have an interest in AI.

R3: Thank you for the suggestions. We agree that it would be ideal to have a representative sample. Randomly selecting from a register is not possible due to The General Data Protection Regulation (a European Union regulation on information privacy in the European Union and the European Economic Area). This does not allow us to have personal information about potential participants (eg contact information, or their clinical interests) until they have consented to the study. To overcome this, recruitment to the study will be through advertisement through national networks (Royal College of General Practitioners, Royal College of Nursing, The Association for Respiratory Technology and Physiology accreditation programme), regional networks (Integrated Care Boards), professional networks in primary care, the institution's website and social media. By advertising and recruiting nationally, the recruitment strategy aims to engage with clinicians from diverse backgrounds in terms of the patient population they serve, age, years of clinical experience, job role and ethnicity. We will offer financial reimbursement for all potential participants participating and completing the study, which may attract some who might not be interested in AI or respiratory diagnostics. This is clarified in the revised manuscript (page 6, paragraph 1).

To account for potential factors that might influence spirometry interpretation, randomisation will include minimization for job role (GP vs not GP) and baseline expertise in spirometry (as measured by whether the participant was accredited in spirometry by the Association for Respiratory Technology and Physiology (ARTP)).

The demographic data will be reported in the manuscript that presents the study results, and therefore the reader will be able to consider 1) whether the population is representative, and 2) whether there were any significant differences in demographics between the intervention and control groups.

C4: It would be useful for the GPs who don't have access in the first round to use it in the second phase and take their feedback

R4: Thank you for the comment. The reviewer is suggesting a "before and after" analysis which is one way of assessing the effects of a digital intervention. However we preferred a two independent sample RCT design that avoids some of the limitations of before and after studies, or cross-over studies. In particular, before and after studies cannot rule out that something other than the intervention may have caused a change, such as spontaneous improvement. Other limitations of before and after studies include learning effects and regression to the mean.

Reviewer: 2

C5: The SPIRO-AID study is an interesting and important study to assess the effectiveness of an AI-intervention to primary care clinicians' assessments of spirometry. The study is well-designed with clearly explained rationale.

R5: Thank you for the positive comments.

C6: My major questions relate to determinations of diagnosis, which is the key outcome being assessed here, given the existence of varying standards, importance of other PFT metrics, and importance of clinical history in making a diagnosis.

R6: Thank you. Determination of the Reference Diagnosis was performed independently by two pulmonologists from a specialist respiratory hospital. They had full access to primary and secondary care medical records and medical history for each of the 50 patient cases. Other than the spirometry report, they also had access to the reports of any investigations performed in primary or secondary care including full lung function tests, imaging (xray, CT scans), cardiac investigations and blood panels.

In contrast, the study participants will be provided with spirometry report only with a one-line clinical

case history comprising age, gender, predominant respiratory symptom and smoking status. This is to replicate real world primary care practice in the United Kingdom where spirometry might be available, but certainly not full lung function data such as gas transfer or lung volumes measured by body plethysmography. An example of the information to be provided to the study participants is shown in Figure 1 of the manuscript.

C7: Do all reports only include spirometry (with no lung volumes or lung capacity), and do they only include pre-bronchodilator spirometry? Spirometry alone is already limited in its diagnostic ability. ILD is not diagnosed based on spirometry alone (and even 'restriction' is best defined based on lung volumes). COPD is preferably diagnosed on post-bronchodilator spirometry. Further, differentiating COPD and asthma would be assisted by post-bronchodilator testing or clinical history.

R7: Please see response R6. Only spirometry data was provided to the trial participants. The spirometry cases were taken from a real-world primary care spirometry service. This service was not commissioned to perform pre- and post-bronchodilator measurements, and therefore only baseline spirometry performed at one timepoint was provided. We recognise that this limited data will not provide the study participants with all the information to make an accurate diagnosis. Indeed, previous studies have shown that experts with information from a medical history and spirometry data alone are correct in 61 % of cases (Decramer et al 2013 [https://doi.org/10.1016/S2213-2600\(13\)70184-X](https://doi.org/10.1016/S2213-2600(13)70184-X)). The planned study is not a diagnostic accuracy study but rather a clinical trial to see whether the use of an artificial intelligence powered software could improve the spirometry interpretation performance of primary care practitioners. Spirometry interpretation performance comprises three main components: diagnostic prediction, assessment of the technical quality of spirometry, and spirometry pattern recognition. The reviewer will note that we powered the study to demonstrate a 10% improvement in correct diagnostic prediction from 55% to 65% of cases.

C8: Along the same lines, what clinical history or context is provided in the cases to participants? Differentiating between many of the diagnoses frequently relies on other clinical history, including symptoms, risk factors, and radiology.

R8: Thank you for this comment. Please see our responses R6 and R7.

C9: Were the two pulmonologists who determined the diagnostic reference standard directed to use any specific criteria? Most pertinently given the focus on COPD, there are varying recommendations on whether to use the FEV1/FVC criteria below the lower limit of normal, or below 70% predicted.

R9: Thank you for this query. Please see Response R6. Both pulmonologists are highly experienced physicians working at a specialist respiratory hospital. Both have expertise in COPD and are aware of the pros and cons of using fixed cut-points versus those based on z-scores in the identification of airways obstruction. They received no external direction and were left at their own discretion to independently provide the most likely diagnosis based on the information from primary and secondary care medical records, and the results of available investigations.

C10: Additionally for the determination of the diagnostic reference standard, is there any assessment of either inter-rater reliability or how pulmonologist interpretations match the AI-generated interpretations?

R10: Thank you. The agreement between pulmonologists for the whole patient cohort (comprising 1121 cases), and how the AI software compared with the reference diagnosis generated by the pulmonologists are the subject of a separate manuscript that is currently under review at another journal. For the 50 patient spirometry sessions used in the online assessment, there was independent agreement between the pulmonologists in 49 cases (95%). In the remaining patient case, there was consensus agreement between the two pulmonologists over the final diagnosis after case discussion.

C11: Page 15, line 35 – Will any subgroup analysis be done of the other specified conditions?

R11: Thank you for this comment. Our planned subgroup analysis focuses on cases of COPD. As the reviewer previously observed, spirometry in primary care is most relevant and important for COPD diagnosis. Spirometry alone is less useful in the diagnosis of the other disease categories.

C12: Page 15, line 41 – Is there any specific reasoning for 70%? Are there any additional planned analyses in which those with <70% completion would be included?

R12: Thank you. The 70% cut-off is arbitrary, but is considered an acceptable level of missing data after discussion amongst the research team, including the study statistician and patient / public involvement representatives. We will compare the baseline characteristics and demographics of those who completed < 70% of questions with those who completed ≥ 70% of questions to identify any systematic differences between completers and non-completers. This has been clarified in the manuscript (page 15, paragraph 4).

Reviewer: 3

C13: It is a well designed study. This study has the potential to serve as an important link to the future where the interpretation of spirometry may be totally automated based on AI. The challenge will be to ensure that clinicians who volunteer to participate in the study do not upgrade their existing understanding regarding spirometry as that will lead to the study bias. Broad advertisement for diverse participant enrolment would be needed. The study design is meticulous and covers all required aspects.

R13: Thank you for the positive comments. Please see response R3 for our planned recruitment strategy.

VERSION 2 – REVIEW

REVIEWER	Brems , J Henry Johns Hopkins University, Department of Pulmonary and Critical Care Medicine
REVIEW RETURNED	17-May-2024

GENERAL COMMENTS	I thank the reviewers for their thoughtful and thorough responses. I have one minor follow-up comment for consideration, and otherwise consider this is in excellent shape for publication. 1) I would consider briefly reporting some metric of agreement between the AI and the pulmonologists, if allowable given their separate pending manuscript. This may be critical to ultimately interpreting the data and understanding how the intervention relates to any between-group difference, or lack-thereof. For example, if the AI has poor agreement with the pulmonologists, then it could mean: i) that the AI model itself is not the cause of the between-group difference (if there are positive findings), or ii) that a better AI model is needed, rather than the notion that AI-aided interpretation isn't effective (if the findings are neutral).
---

VERSION 2 – AUTHOR RESPONSE

Reviewer: 2

Comments to the Author:

I thank the reviewers for their thoughtful and thorough responses. I have one minor follow-up comment for consideration, and otherwise consider this is in excellent shape for publication.

1) I would consider briefly reporting some metric of agreement between the AI and the pulmonologists, if allowable given their separate pending manuscript. This may be critical to ultimately interpreting the data and understanding how the intervention relates to any between-group difference, or lack-thereof. For example, if the AI has poor agreement with the pulmonologists, then it could mean: i) that the AI model itself is not the cause of the between-group difference (if there are positive findings), or ii) that a better AI model is needed, rather than the notion that AI-aided interpretation isn't effective (if the findings are neutral).

Response:

Thank you for the comment. We agree this is an important point and will report the diagnostic prediction performance of the AI software alone compared to the reference diagnosis produced by the pulmonologists (added to manuscript, page 12).